# Influence of Photodynamic Therapy on Lichen Sclerosus with Neoplastic Background

**DOI:** 10.3390/jcm11041100

**Published:** 2022-02-19

**Authors:** Magdalena Bizoń, Danuta Maślińska, Włodzimierz Sawicki

**Affiliations:** 1Chair and Department of Obstetrics, Gynecology and Gynecological Oncology, Medical University of Warsaw, 02-091 Warszawa, Poland; saw55@wp.pl; 2Department of General and Experimental Pathology, Medical University of Warsaw, 02-091 Warszawa, Poland; danuta.maslinska@wum.edu.pl

**Keywords:** lichen sclerosus, photodynamic therapy, vulvar disease

## Abstract

Background: Lichen sclerosus is the most common nonmalignant vulvar disease with morbidity in postmenopausal age. The first line of treatment is corticosteroid therapy. In case of insufficiency, tacrolimus or pimecrolimus can be provided. Photodynamic therapy (PDT) can be used as alternative way of treatment while symptoms recurrent despite other methods. Methods: the analyzed population of 182 women with diagnosis of lichen sclerosus treated using PDT was divided into three groups: patients with neoplastic disease or intraepithelial neoplasia; those with a positive family history of neoplastic disease; and a control group with no neoplastic disease and no familial history of neoplastic diseases. Results: Reduction of vulvar changes was assessed in the whole vulva in the groups as 21.9%, 21.2% and 21.8%, respectively. The most frequent symptom, itching, was reported to decrease in all groups, 39.3%, 35.5% and 42.5%, respectively. Improvement of quality of life was assessed in 91.3% of the whole group, stabilization of lichen sclerosus in 7.1% and progression in 1.6%. Conclusions: Photodynamic therapy gives positive results in most cases. Improvement after PDT is observed in objective vulvoscopic assessment and in subjective patients’ opinions. Neoplastic disease in the past can influence the effectiveness of PDT.

## 1. Introduction

In postmenopausal women, the most frequent cause of itching and burning is vulvovaginal atrophy. Vulvar changes can also occur due to allergic contact with dermocosmetics. Differential diagnosis is based on autoimmunological or neoplastic etiology [1].

Biopsy of vulva establishes the final diagnosis of dermatosis. The most frequent histological recognition is vulvar lichen sclerosus [2]. Symptoms are tiring and cumbersome for patients and reduce quality of life. The aim of the treatment is to reduce symptoms and elevate well-being. The most significant target is to attain the longest time free of progression of lichen sclerosus.

The frequency of lichen sclerosus is estimated at 1 per 30 to 1 per 1000 persons. One of the reasons of vulvar lichen sclerosus is decreasing level of estrogens, which is mostly observed in childhood and in menopausal age. However, vulvar lichen sclerosus can appear in every period of life. The mean age of diagnosis is described between 52.6 and 60 years [3]. Micheletti et al. in retrospective study presented population of 976 cases of lichen sclerosus with mean age of 60 years, but with range 8 to 91 years [4]. Lee et al. in prospective cohort study observed women with lichen sclerosus also in reproductive age where the youngest patient was 18 [5]. Moreover, among children the percentage of boys with this diagnosis is higher than girls. Statistics show a rate of 1 per 900 girls up to 16 years old and 1 per 200 boys up to 14 years old [6]. Lichen sclerosus is presented commonly in cases of prepubertal girls with an average age of 7.6 years and in peri- and postmenopausal patients with a mean age of 52.6 years [7].

Lichen sclerosus is mainly located in the anogenital area in 85 to 98% of cases. Extragenital localization is observed in 15–20% of patients regardless of gender [8]. There was no observed malignant transformation in changes presented in extragenital area [9].

Etiology of lichen sclerosus is multivariate. Familial background was documented in 12% of patients [10]. Familial lichen sclerosus correlates with presence of anti-TPO antibodies and HLA-B*15-DRB1*04-DRB4*. It has been shown that HLA DQ8 and DQ9 coexist with lichen sclerosus presented in men and women. HLA DR11 and DR12 more frequently appear in male lichen sclerosus [11]. Co-existence of HLA antigens and antithyroid antibodies suggests co-incidence of lichen sclerosus and autoimmunological diseases. Immunological diagnostics should be extended especially in the first line of the family of patients with lichen sclerosus [12].

Etiology of lichen sclerosus is correlated with previous *Chlamydia trachomatis* infection. Antibodies IgM and IgG of *Chlamydia trachomatis* were observed in patients with diagnosis of lichen sclerosus. *Chlamydia trachomatis* antigens were observed in 12% of women with lichen sclerosus and 20% with diagnosis of vulvar cancer [13]. In 39% of vulvar lichen sclerosus, ANA antibodies were also detected [14]. A correlation between positive *Borrelia burgdorferi* antibodies and lichen sclerosus was reported [15]. Moreover, Epstein-Barr virus (EBV) was discovered in vulvar biopsies of patients with lichen sclerosus [11].

A higher percentage of HPV infection in lichen sclerosus was detected in males (29%) than females (8%) [16]. In a metanalysis, a correlation between infection with HPV and vulvar intraepithelial neoplasia (VIN) was observed in 84% cases. Neoplastic transformation was claimed in 10% of patients [17]. Undifferentiated VIN did not correlate with lichen sclerosus, but most frequently vulvar squamous cancer is observed [18]. However, lichen sclerosus was identified in the area of differentiated VIN. Because of the high risk of neoplastic transformation, every new change should be biopsied and histologically verified [19].

Statistics show that the frequency of transformation of lichen sclerosus into vulvar cancer is circa 3–6%. In 60% of histological samples of vulvar cancer, lichen sclerosus is detected in the area of margins [20]. It was observed that overexpression of a marker of angiogenesis (VEGF) and COX-2 is associated with transformation of vulvar cancer [21]. There is also a correlation between hypermethylation of *CDKN2A* with vulvar squamous cell carcinoma and VIN in comparison to lichen sclerosus [5].

On the other hand, other localizations of cancer are described with lichen sclerosus. There were published cases of endometrial cancer and lichen sclerosus [22]. Symptoms appear months, sometimes years, after diagnosis of this disease [23]. Postoperative radiotherapy in endometrial cancer can also develop vulvar neoplasia, which can transform more frequently into vulvar cancer [24,25].

Treatment of lichen sclerosus is based on ointments. Everyday use reduces severe, irritable symptoms. In the case of progression of the disease, local steroid therapy is used. The first line of treatment recommended for active lichen sclerosus are potent corticosteroids administered for 12 weeks [26]. The most common adverse effects observed during longitudinal topical corticosteroids therapy is atrophy and dermal thinning [27]. Ineffectiveness of the standard method of treatment spurs the search for alternative methods. Benefits are observed from using photodynamic therapy. Photodynamic therapy is indicated for lichen sclerosus with no results after pharmacological treatment. The dual mechanism of photodynamic therapy depends on the overactive oxygen molecule, which arrests the inflammatory process in lichen sclerosus. Reactivity of tissue depends on epigenetic background, so light energy absorption can influence this in different ways. Photosensitizers using before PDT caused damage of inflamation process and fibrosis limited to healthy tissue. In lichen sclerosus there are no proven advantages of surgical treatment [28].

The aim of this study was to assess the effectiveness of photodynamic therapy in patients with a diagnosis of lichen sclerosus and concomitant neoplastic disease or a family history of cancer.

## 2. Materials and Methods

In the Outpatient Clinic, 246 patients with histological diagnosis of lichen sclerosus were observed. Lichen sclerosus is the most common nonmalignant vulvar dermatosis. The analysis was based on 182 patients with this diagnosis according to accompanying diseases. Average age of patients is circa 63 years (62.92 ± 12.33). Onset of vulvar symptoms of lichen sclerosus is at the age of circa 53 years (53.12 ± 13.76). Minimum age was 29 years and maximum 88 years. Remaining patients were under observation with no symptoms or were treated only by pharmacological methods. Most of the analyzed group (78%) presented to the gynecologist for the first time because of vulvar symptoms in postmenopausal age. Average time of the visit was circa 7 months (7.01 ± 6.15) after the onset of vulvar complaints. Inclusion and exclusion criteria are included in Table 1.

Photodynamic therapy is an alone method of treatment. Patients were qualified for 10 courses of PDT every week, which stands for one cycle of treatment. Every course lasted 10 min. A total of 2 hours before therapy, photosensitizer—δ-aminolevulinic acid is applicated on vulvar skin. Thanks to the photosensitizer, vulvar skin was more reactive for light from the photodynamic lamp. The cycle of photodynamic therapy can be repeated due to recurrence of vulvar symptoms. PDT was performed using the PhotoDyn501 with emission of light at 630 nm wavelength and power density of 204 mW/cm^2^.

The analyzed population was divided into 3 groups. Group A included 44 patients (24.1%) with neoplastic disease or vulvar (VIN), cervical (CIN) or endometrial (EIN) intraepithelial neoplasia. The percentage distribution of different kinds of neoplastic disease and intraepithelial neoplasia is presented in Figure 1. Mean age was 64.57 ± 11.58 years. Group B included 51 women (28.1% of the total population) with a family history of neoplastic disorders in the first line (father–mother and grandfather–grandmother) and second line, e.g., brother or sister, with mean age 63.13 ± 11.3 years. Group C was the control group with 87 patients (47.8%) with no neoplastic diseases and no familial history. Mean age of group C was 62.44 ± 12.89 years.

Analysis was based on objective vulvar symptoms assessed by an expert. Analysis of objective symptoms of vulvar disease was based on a special scale of our own authorship to evaluate features of lichen sclerosus.

The first scale was based on different types of features of vulvar skin assessed in vulvoscopy [Table 2]. The second one concerned the extent of vulvar changes and their intensity [Table 3]. Points are collected according to presence of change, its intensity and extent. The larger and more advanced the dermatosis is, the more points are given. The assessment was done before the onset of photodynamic therapy and after completing of treatment. The sum of points given during vulvoscopy were then compared. Results showed benefits and disadvantages of photodynamic therapy.

To obtain the total assessment of vulvoscopic improvement, points were counted all together before and after 10 courses of photodynamic therapy. A decreasing number of points was considered improvement through therapy, because all changes disappeared or declined. Stabilization of lichen sclerosus is the state when vulvar appearance was the same before and after the treatment and the sum of points was balanced. A higher level of points after photodynamic therapy indicates progression of lichen sclerosus.

A questionnaire containing questions for patients concerning symptoms of lichen sclerosus and its intensity was used.

The most frequent symptoms of lichen sclerosus are itching, burning and pain. The intensity of complaints influences the quality of everyday life. The stronger the symptoms are, the more quality of life decreases. A special questionnaire based on a 10-degree value of intensity of every symptom was used. Patients from every group assessed the level of itching, burning and pain before initiation of photodynamic therapy and after the treatment. The scale consists of 10 degrees, where 0 means no symptoms and 10 means maximal intensity of physical conditions. Effectiveness of PDT in vulvar itching was measured as the difference between intensity of conditions before and after treatment.

Quality of life is a difficult value to estimate. It is a subjective opinion of every woman. Standard of life depends on age, physical condition, activity and habits. These factors influence the level of quality of life. To compare comfort of life after ending PDT, a special questionnaire was created on our own authorship. Every woman has to consider the effectiveness of PDT as a percentage. The scale is described in Table 4. We decided to create few questions contains effectiveness of photodynamic therapy, which was easy and short time taken.

All results were statistically analyzed using the parametric Student’s *t*-test and non-parametric Mann–Whitney U test and Fisher’s test. For correlation of two values the chi-square test was used. For all tests a value of p less than 0.05 was considered significant.

The study protocol was approved by the local ethical committee (Nr AKBE/85/2018).

## 3. Results

### 3.1. Clinical impact of Photodynamic Therapy

#### 3.1.1. Objective Benefits of Photodynamic Therapy

Lichen sclerosus occupied all the vulva before photodynamic therapy was observed in every third patient of the whole analyzed population (47 patients, 26%). The inguinal area was extended in 14 women (7.6%) and anorectal localization of lichen sclerosus was detected in 32 patients (17.6%). In other cases, lichen sclerosus was limited to the urethra (1.3%), clitoris (3.9%), and labia minora (11.7%) and labia majora (18.5%).

In all groups, after the completion of photodynamic therapy, the regression of lichen sclerosus changes was observed. The largest improvement of lichen sclerosus after PDT occurred in patients in whom the whole vulva was occupied. In group A it was 21.9% of cases, in group B 21.2% and group C 21.8%.

The reduction of dermatosis in the anorectal area was noted more in patients from group C (14.5%) than in group B (9.9%) and group A (6.2%).

The smallest improvement in the inguinal area was 2.8% from group A and C. There was no improvement in group B (Table 5).

The lowest percentage of progression of lichen sclerosus was observed in group C with no neoplastic disease and no familial history (11.4%). Progression of lichen sclerosus occurred in 17.6% of patients from group B and 18.2% from group A (*p* < 0.05).

Benefits of PDT were described in treatment of patients from group B (56.9%) and from group C (55.3%). A worse response to photodynamic therapy was observed in group A—47.7% women with improvement after vulvoscopic assessment (*p* < 0.05).

Lack of effects after treatment, which means stabilization of lichen sclerosus, was comparable in all groups, with percentages of 34.1%, 25.5% and 33.3% (*p* < 0.05).

Average points for intensity of lichen sclerosus before onset of PDT according to the scale were assessed as 3.3 in group A; 3.2 in group B; 3.2 in group C. What is more, specific values were observed in subgroups of group A. Average points for intensity of lichen sclerosus in patients with vulvar cancer in the past were assessed as 3.5, while in patients with VIN—3.9 (*p* < 0.05).

The most significant difference in morphology of the vulva after the end of PDT was noted in group C with average points—1.8; then in group B—2.5 and group A—2.7. However, average points after the end of PDT were 2.9 for vulvar cancer and 2.3 for VIN. The reason for this phenomenon is better predisposition and absorption of light energy on skin with no neoplastic background (Figure 2) (*p* < 0.05).

No progression of lichen sclerosus to vulvar cancer was observed after photodynamic therapy. There was also no progression to VIN.

No primary hypersensitivity was observed after PDT. Patients did not complain of pain, itching and any discomfort in vulvar area after therapy.

#### 3.1.2. Subjective Benefits of PDT

The most often-reported symptom is itching. Before onset of PDT, this symptom was present in every group, with percentages of 76.2%, 71.7% and 66.9%. The highest level of intensity of itching was observed in women with neoplastic disease in the past. In group A, vulvar itching decreased and was reported by 39.3% of patients, in group B—35.5%, group C—42.5%. In subgroups of group A, intensity of itching was observed in 45% of patients with vulvar cancer and 38.4% with VIN (*p* < 0.05) [Figure 3].

Vulvar burning was the second most common complaint. Before PDT, it was reported by 57.8% of women from group A (75.2% women with vulvar cancer and 54.6% with VIN), 49.2% from group B and 55.8% from group C. After treatment, only 27.8% of patients from group A (50.1% with vulvar cancer and 27.3% with VIN), 23.2% from group B and 18.7% from group C complained of burning (*p* < 0.05) [Figure 4].

The highest level of improvement of vulvar symptoms was observed in patients with no familial history and no neoplastic disease (37.1%). In other groups it was noted as 30% from group A and 26% from group B (*p* < 0.05).

Vulvar pain manifested in 33.1% of women from group A (87.9% of patients with vulvar cancer and 33.4% with VIN), 24.2% from group B and 19.8% from group C. After PDT, vulvar pain decreased in 17.2% of patients from group A (43.8% with vulvar cancer and 11.1% with VIN), 10.1% from group B and 10.9% from group C (*p* < 0.05) [Figure 5].

#### 3.1.3. Quality of Life

Symptoms totally disappeared in 14.8% of cases. A 70% improvement of quality of life was reported by 30.8%. Every third patient (35.2%) noted a 50% decrease of symptoms. In total, 3.4% and 7.1% of women claimed that the level of improvement was 30% and 10%, respectively. No improvement was reported by 7.1% of patients. Only 1.6% reported progression of lichen sclerosus and worse quality of life (Figure 6) (*p* < 0.05).

## 4. Discussion

PDT can be an alternative method of treatment of nonmalignant vulvar dermatoses in cases of ineffective ointment treatment. Investigations have revealed a high percentage of satisfaction after this kind of treatment. Patients were qualified to our study with recurrence of vulvar symptoms more than 3 months after topical steroid treatment because of willingness of recognizing a new method of treatment. Our results are similar to those reported by other authors [29,30,31].

The analyzed group is representative because all the patients are in perimenopausal and postmenopausal age, which is the time of peak incidence of lichen sclerosus [32]. Renaud et al. analyzed a similar group of women. They found a worse response to local glucocorticosteroid therapy in the older group of patients above 70 years [33].

According to current recommendations, the first line of treatment was local glucocorticosteroid therapy [34]. Recurrence of vulvar symptoms is an indication for the second-line therapy. If recurrent steroid therapy is insufficient, it is possible to use an alternative method. One such off-label treatment is photodynamic therapy [35,36]. Vulvar symptoms are so irritable for the patients that they decrease quality of life. Searching for a new method of treatment is necessary because of theinsufficiency of local conservative therapy. Prodromidou et al. reviewed 11 studies showing promising effects in lichen sclerosus diagnosed women [37]. Li et al. revealed an effective rate of PDT in 92.31% cases using 5-aminolevulinic acid. Influence of PDT was mildly toxic for most patients [38].

Lower concentration of estrogens in postmenopausal age predisposes to morbidity of lichen sclerosus. Similar conditions are presented by women after bilateral adnexectomy or removal of an ovary in patients of reproductive age. Lack of endogenous estrogens caused artificial menopause, which can introduce lichen sclerosus [39]. Lagerstedt et al. detected expression of estrogens receptors in lichen sclerosus. Lower expression of ERRα receptors located in cytoplasm was observed in vulvar lichen sclerosus, vulvar intraepithelial neoplasia and vulvar cancer compared to healthy skin. There was no difference in the expression of EERβ and EERγ receptors. Estrogen receptors were present in the vagina. On the other hand, in the vulva more androgen receptors were detected. Expression of estrogen receptors was correlated with age and hormonal status of the patient [40]. Higgins et al. observed progression of symptoms in the anogenital area in patients with lichen sclerosus after menopause and bilateral adnexectomy [41]. In the analyzed population, less than half of the women (45.1%) had undergone the operation in the past. Just 13.6% of this group underwent hysterectomy with bilateral adnexectomy, 31.5% bilateral adnexectomy and 9.9% urogynecological operations because of urinary incontinence.

The analyzed population in our research included 3.3% of patients after vulvectomy because of vulvar cancer and VIN accompanied by lichen sclerosus. In these cases, ineffective therapy based on glucocorticosteroids and progression of symptoms were indications for PDT. Soergel et al. also used PDT in those cases [42].

Tribbia et al. described coexistence of vulvar lichen sclerosus and vulvar cancer defined in 59% of cases. Presence of lichen sclerosus and VIN 3 was discovered in 35% of cases [43]. In the analyzed population, the presence of lichen sclerosus and vulvar cancer was diagnosed in 2.2% of patients and lichen sclerosus with VIN in 8.2% of women. Hart et al. detected the coexistence of lichen sclerosus with endometrial cancer (6.5%), cervical cancer (1.1%) and colon cancer (1.1%) in a population of 92 women [44]. Epidemiological data of our research were appropriately 1.6%, 0.6% and 1.1%, respectively.

Moreover, Hanonen et al. conducted a study from 1970–2014 on a population of 7616 patients with diagnosis of vulvar lichen sclerosus. They observed 812 women (10.7%) with vulvar lichen sclerosus and concomitant neoplastic disease: breast cancer (20.9%), cervical cancer (0%), endometrial cancer (5.1%), ovarian cancer (2.6%), vaginal cancer (0.5%) and vulvar cancer (22.4%) [45]. In our research, the percentages of co-occurring diseases were similar: breast cancer (18.2%), cervical cancer (2.3%), endometrial cancer (6.8%), ovarian cancer (2.3%), vaginal cancer (0%) and vulvar cancer (9.1%). Coincidence of symptoms of the anogenital area and breast cancer correlated with lower concentration of estrogen was described by Mac Bride et al. These symptoms should also be differentiated from lichen sclerosus [46].

Characterization of subgroups was conducted due to presence of lichen sclerosus and family history of neoplastic disease. This predisposition is correlated with HLA haplotype, which was documented in the literature [47]. Familial background is also observed in neoplastic diseases.

Vulvoscopic assessment before and after PDT was used to compare clinical features of lichen sclerosus. In advanced stages of lichen sclerosus, atrophy of the labia, narrowing urethra area, fibrosis of the clitoris area and shrinking vagina are observed. Extragenital localization is observed in 20% of patients. It can be alone with no coexistence in the genital area and can be asymptomatic [48].

Efficacy of PDT was measured by comparison of vulva morphology before and after PDT. Mazdziarz et al. presented results of patients with diagnosis of lichen sclerosus treated with PDT where a 0–5 scale was used to assess improvement of symptoms after PDT [49].

Lichen sclerosus also affected the inguinal area. Before onset of PDT this localization was noted in 6.3% of group A, 5.6% of group B and 9.1% of group C. Regression of changes in this area was reported only in patients from group C in 6.3% of cases. To our knowledge, there is no other publication about inguinal localization of lichen sclerosus.

Typical localization of lichen sclerosus is the anogenital area. Criscuolo et al. described benefits of PDT especially in perianal lichen sclerosus [50]. Similar results were obtained in our research. The perianal area is affected in 15.6% of patients with neoplastic disease, 12.7% with familial history and 23.6% without familial history. Decreasing changes of disease after completion of PDT is observed in 14.5% of group C, 9.9% of group B, and least in group A. It is believed that some localizations are more predisposed to regression after this kind of treatment. Liu et al. conducted a prospective study based on clinical and dermoscopic assessment of vulvar lichen sclerosus after 5-aminolevulinic acid PDT. After sixth course of PDT, they observed increasing score of vessels and decreasing score of bright white and white-yellowish area of early changes of lichen sclerosus. Vulvoscopy is significant for assessment of effectiveness of PDT [51].

All results revealed improvement after PDT, which is more spectacular in patients with no burden than with neoplastic disease or familial history. PDT gives less benefits in the group with positive familial neoplastic disease history. There has been no investigation comparing groups with lichen sclerosus with a neoplastic background.

Quality of life is a measure of well-being. The best method to estimate quality of life is a subjective questionnaire. Symptoms and discomfort in the anogenital area caused by lichen sclerosus significantly reduce the quality of life. The aim of treatment of lichen sclerosus is to reduce vulvar symptoms assessed only by patients. Lansdorp et al. used a special scale called Skindex-29 in nonmalignant dermatoses with information about symptoms, feelings, sexual life, fatigue and influence on everyday life. In this research the SF-12 scale applied to women from Denmark with lichen sclerosus was also used to assess stage of disease and subjective feelings [6]. The results were similar to those in our investigation.

Sheinis et al. in a metanalysis compared different scales assessing features of lichen sclerosus based mainly on symptoms, clinical changes and histological aspects. The Female Sexual Function Index and the Dermatology Life Quality Index were also used to assess quality of life among patients [52]. In our research a special questionnaire was used. It was a 10-degree scale, where 0 means no symptoms and 10 means maximal severity of the problem. Questions concerned the most frequent symptoms, such as itching, burning and pain. Gradation of the scale was similar to the VAS scale. In the literature, different scales have been used [13,53,54]. Knowledge of gradation of symptoms help to measure level of quality of life and find a method of treatment with higher efficiency. In our research, patients had the possibility to compare feelings after ending PDT. Thanks to the special scale, women can choose the level of improvement. Overall, at least 50% improvement was reported in 80.8% of cases. Mazdziarz et al. observed 77.5% of positive responses to treatment. Moreover, in our research quality of life after PDT showed improvement in 87.25% of cases [13,37,49,55].

## 5. Conclusions

To sum up, it is possible that neoplastic disease in the past can influence the effectiveness of PDT. There are few publications available about the problem of PDT in lichen sclerosus with a neoplastic background. Nowadays, morbidity of neoplastic diseases is elevated, so morbidity of lichen sclerosus may also increase. Further investigations are necessary to identify these patients and provide an individual method of treatment to prolong life without recurrent symptoms.

## Figures and Tables

**Figure 1 jcm-11-01100-f001:**
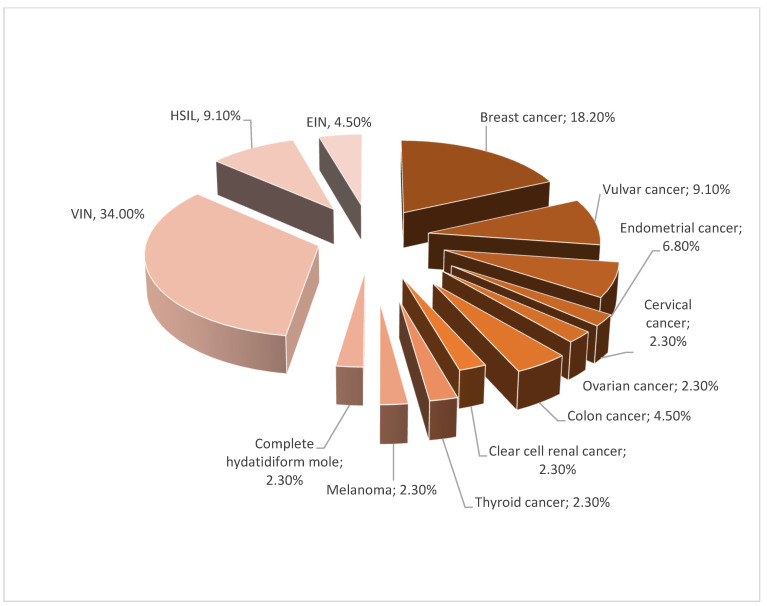
Segregation of group A.

**Figure 2 jcm-11-01100-f002:**
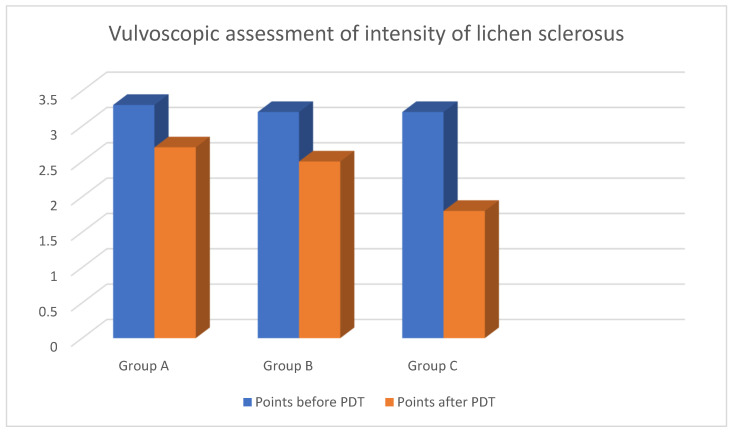
Vulvoscopic assessment of intensity of lichen sclerosus.

**Figure 3 jcm-11-01100-f003:**
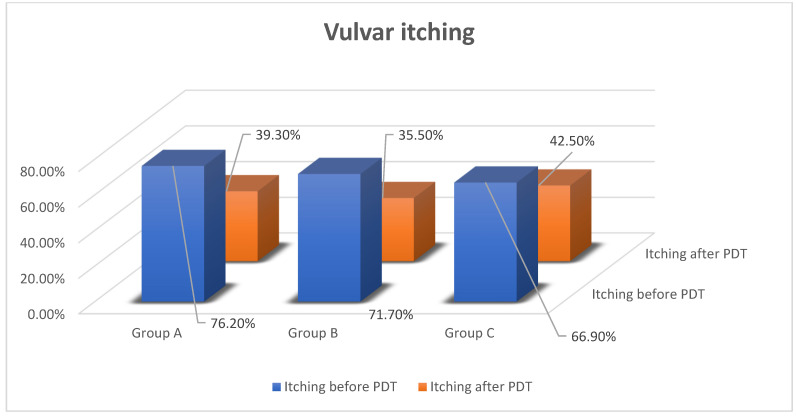
Effectiveness of PDT in vulvar itching.

**Figure 4 jcm-11-01100-f004:**
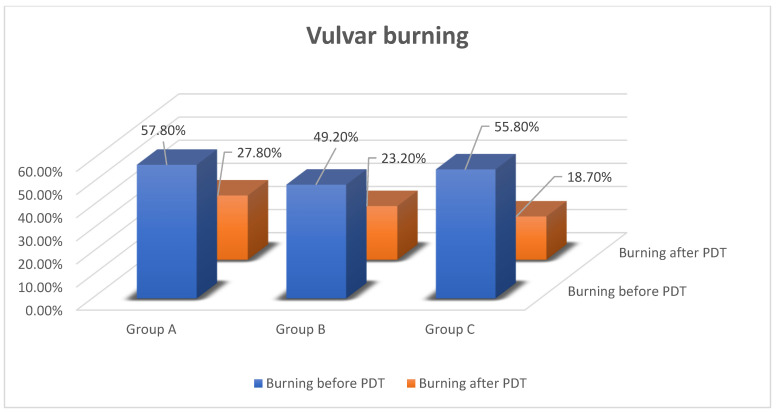
Effectiveness of PDT in vulvar burning.

**Figure 5 jcm-11-01100-f005:**
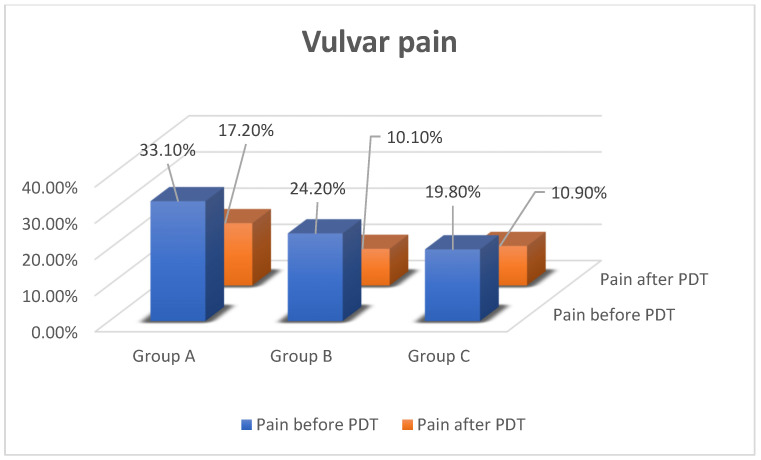
Effectiveness of PDT in vulvar pain.

**Figure 6 jcm-11-01100-f006:**
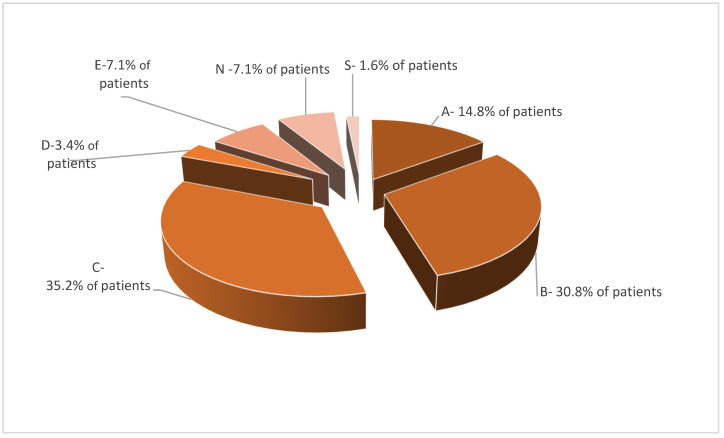
Satisfaction with photodynamic therapy in patients’ opinion.

**Table 1 jcm-11-01100-t001:** Inclusion and exclusion criteria.

Inclusion Criteria		Exclusion Criteria
1.	Lichen sclerosus histopathologically confirmed	Histological diagnosis not including lichen sclerosus
2.	Recurrence of vulvar symptoms more than 3 months after locally conservative therapy: ▪ Glucocorticosteroidotherapy ▪ Tacrolimus ▪ Ointments	2. Good response to conservative therapy
3.	No use of PDT before	3. PDT used before as alternative therapy
4.	Age above 18 years	4. Age below 18 years
5.	Conscious agreement for using PDT	5. Lack of conscious agreement for using PDT
6.	Previous diagnosis of neoplastic disease or neoplasia	6. Neoplastic diseases or neoplasia diagnosed after histologically confirmation of vulvar lichen sclerosus

**Table 2 jcm-11-01100-t002:** Features of vulva.

	Features	Points
1.	Vulva with no macroscopic changes	0
2.	Reddening	
	Present	1
	Absent	0
3.	Atrophy	
	Narrowing of atrium of vagina	1
	Labia asymmetry	1
	Low degree of atrophy (labia shrunk less than 1/2)	2
	Medium degree of atrophy (labia shrunk more than 1/2)	3
	High degree of atrophy (atrophy of all vulva)	4
4.	Leukoplakia	
	Absent	0
	Single focus of leukoplakia	1
	Low degree (less than ½ of vulva)	2
	Medium degree (more than ½ of vulva)	3
	High degree (all vulva affected)	4
	All vulva + groin affected	5
5.	Erosion	
	Absent	0
	Low degree (less than ½ of vulva)	1
	High degree (more than ½ of vulva)	2
6.	Ulcers	
	Absent	0
	Low degree (less than ½ of vulva)	1
	High degree (more than ½ of vulva)	2
7.	Hyperkeratosis	
	Absent	0
	Less than ½ of vulva	1
	More than ½ of vulva	2
8.	Ruptures	
	Absent	0
	Present	1
9.	Excoriation	
	Absent	0
	Present	1
10.	Subcutaneous hemorrhages	
	Absent	0
	Present	2
11.	Wounds	
	Absent	0
	Single presentation	3
	Numerous	5
12.	Edema of vulva	
	Absent	0
	Present	3

**Table 3 jcm-11-01100-t003:** Localizations of vulvar changes in lichen sclerosus.

	Localizations of Vulvar Changes in Lichen Sclerosus	Points
1.	Vulva with no macroscopic changes	0
2.	Localization of urethra	
	Present	1
	Absent	0
3.	Clitoris	
	Present	1
	Absent	0
4.	Vulvar vestibule	
	Present	1
	Absent	0
5.	Labia minora	
	Both-sided	2
	One-sided	1
	Absent	0
6.	Labia majora	
	Both-sided	2
	One-sided	1
	Absent	0
7.	Perianal localization	
	Present	1
	Absent	0
8.	Inguinal localization	
	Both-sided	2
	One-sided	1
	Absent	0
9.	Totally affected vulva	
	Present	10
	Absent	0

**Table 4 jcm-11-01100-t004:** Level of satisfaction of patients after PDT.

Grade	Satisfaction of Patients after PDT
A	70–100% improvement after PDT
B	50–70% improvement after PDT
C	50% improvement after PDT
D	30% improvement after PDT
E	10% improvement after PDT
N	No changes after PDT
S	Symptoms are more intense than before PDT

**Table 5 jcm-11-01100-t005:** Localization of lichen sclerosus before and after PDT.

	Affected Anus before PDT	Affected Anus after PDT	Affected Inguinal Area before PDT	Affected Inguinal Area after PDT	All Vulva Affected before PDT	All Vulva Affected after PDT
Group A	15.6%	9.4%	6.3%	9.1%	31.3%	9.4%
Group B	12.7%	2.8%	5.6%	5.6%	25.4%	4.2%
Group C	23.6%	9.1%	9.1%	6.3%	27.3%	5.5%

## Data Availability

Data avaible in a publicly accesible repository.

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
