# Peer review of "Influence of Photodynamic Therapy on Lichen Sclerosus with Neoplastic Background"

_jcm, 2022, doi:10.3390/jcm11041100_

Round 1
Reviewer 1 Report
The scientific points raised have been well revised, although the minority remains including: i) inadequate citation of the references (page 2, the last paragraph), and ii) English wordings (latin words, etc). These should be corrected properly.
Author Response
Dear Reviewer,
I have tried to do the best I can correction of points you have given me.
Below I have written all changes:
- I have modified text of the last paragraph page 2 and now in my opinion reference is adequate. (line 86-87)
- I have changed some words from latin to English. I hope now it is correctly everything.
I would be grateful for your opinion. I hope, everything will be correct now.
Your sincerely,
Magdalena Bizoń
Reviewer 2 Report
I enjoyed reading your article. Below are my comments:
I believe the lichen sclerosus is overlooked in women of reproductive age and I am concerned that your introduction promotes that belief that it is not seen in that age group. I would like to see you include a statement about it occurring in women of reproductive age as well.
Below are more specific comments:
Line 51 – what is postponed chlamydia infection?
Line 64 -what is planoepitheliale vulvar cancer
Line 84 -85 – what evidence that topical steroids used locally can cause systemic immunosuppression?- although theoretically possible, in reality extremely unlikely
Group A line 115 118 – broad; cervical cancer unlikely related to LS; more likely to hpv; so mixed etiology group?
Line 124 -126 – analysis objective vulvar symptoms – but “special scale of our own authorship” not validated
Line 150-155 – quality of life –“ questionnaire… own authorship” there are validated quality of life tools and for sexual health as well ; why not use those?
197 – no sided effects after PDT – seems unlikely -no pain, no itch no localized discomfort?
250 discussion – are these patients that failed topical steroids or patients that recurred after stopped topical steroids; many recommend continued use of topical steroids to maintain improvement once treated;
Author Response
Dear Reviewer,
I have tried to do the best I can correction of points you have given me.
Below I have written all changes:
- I have modified introduction according to common age of women with diagnosed vulvar lichen sclerosus (lines 34-41)
- Postponed Chlamydia infection – I have modified as previous Chlamydia infection (line 57)
- Planoepitheliale Vulvar cancer – I have modified as vulvar squamous cancer (line 68)
- I have modified adverse effects of topical steroids (lines 86-87)
- According to group A, in every case lichen sclerosus was developed after diagnosis of neoplastic disease. Neoplastic diseases or neoplasia were diagnosed before confirmation of vulvar lichen sclerosus. Exclusion criteria was diagnosis of neoplastic diseases or neoplasia after histologically recognition of vulvar lichen sclerosus. We want to observe response of photodynamic therapy on neoplastic background. Group A is heterogeneous one because of a few cases of neoplastic disease. Further investigations and bigger population is needed to differentiate advantages and disadvantages of photodynamic therapy on vulvar lichen sclerosus developing in patients with neoplastic disease in the past.
- Analysis of objective vulvar symptoms was based on numerous scales of points of clinical symptoms from 0 to 10 given by patient before and after PDT. Results were compared and analyzed.
- Satisfactory improvement was assessed subjectively by patients according to global condition and physical status. Questionnaire was based on asking for percentage of satisfaction after PDT in subjective opinion of patient. Only according to description of degree in special protocole.
- Sides effects after PDT I have modified (lines 200-201)
- Patients with vulvar lichen sclerosus taken part in the study were after topical steroids therapy with recurrent symptoms and would like to introduce new method of therapy. I have provided more information in Discussion (lines 254-259)
I would be grateful for your opinion. I hope, everything will be correct now.
Your sincerely,
Magdalena Bizoń
Round 2
Reviewer 2 Report
Thank you for revising the article which I think is improved.
I would suggest small edits be made to the English to make it read smoother and with more clarity.
This manuscript is a resubmission of an earlier submission. The following is a list of the peer review reports and author responses from that submission.
Round 1
Reviewer 1 Report
This well-written study assesses the effects of photodynamic in women with patients with cancer, those with a positive family history of cancer and with no neoplastic disease and no familial history of cancer. The results of this study are intriguing, but certain methodological aspects have to be more clear.
Comments are included below:
Introduction:
- Authors have to include some phrases about dual pathogenesis of vulvar cancer (HPV-associated and –independent) and that LS is considered to be one of the lesions associated to a second pathway.
- Line 35-37: it is better to refer to men and women, rather than to boys and girls
- Lines 41-46: The authors should include also the role of the HLA DQ7 HLA DQ8 and DQ9 in women with LS (see reference 11)
- Lines 47-52: it would be worth to include that the role of infections is still controversial
- Lines 53-54: The paper referenced as 11 does not include HPV among the etiological factors of LS. Perhaps, the authors referred to EBV virus..The authors have to be especially careful about this aspect, as the majority of evidence supports that LS is mostly found in HPV-negative patients.
- Line 56: expand the abbreviation of VIN
- Line 66: names of the genes have to be written in cursive
- Line 66: this phrase has to be reformulated. What “planoepithelial” vulvar cancer actually means?
- Lines 73-75: this paragraph on the LS in ovary is not relevant and can be omitted.
- Lines73-74: the phrase sounds like LS was found in the ovary; rephrase.
- Lines 76-84: include some references to support evidence displayed.
M&M:
- Lines 100-108: It is not clear why authors include in the group A also patients with cervical or endometrial intraepithelial neoplasia, and those with other than vulvar cancer (breast cancer as well?), while patients with vulvar cancer comprise very low percentage. I understand from the abstract that this work focuses on the vulvar lichen sclerosus.
Results:
- Lines 152-153: specify the percentages of women with LS for each of the locations.
Discussion:
- Better organization of ideas is needed in this section. It seems like a collection of statements not highly related between them.
- Line 225: Not necessary to expand PDT abbreviation, it was previously expanded.
- Line 237 and 259-261: This data should ideally go to the M&M section.
Reviewer 2 Report
To the authors,
Bizon et al. presented a therapeutic efficacy of photodynamic therapy (PDT) to patients with lichen sclerosus (LS), an inflammatory anogenital dermatosis, who were unresponsive to the standard topical therapy. The study design looked simple but indeed meaningful, included a reliable number of subjects (n=182), and finally achieved satisfactory outcome, such as improvement of visual examination and major symptoms, particularly restoring their QOL. However, in my personal concern, there remains insufficient results and presentations to reach the author's consideration, as below.
- In the abstract (background), the description that PDT is usually used sounds overinterpretation. Of course, there have been single case reports and case series demonstrating the efficacy of PDT in LS, but PDT therapy remains yet to be one of adjunctive and optional treatments. Indeed, the author has already toned it down separately, as ‘an off-label treatment’ (Discussion, 3rd paragraph). In addition to this, PDT therapy should be avoided during a topical use of tacrolimus or pimecrolimus, calcinurin inhibitors that may be effective for topical steroid-resistant LS. On the basis, the study limitation also needs to be described comprehensively in the discussion.
- In the introduction (3rd paragraph), the description for the disease susceptibility simply lists the previous data and seems vague, not much of message. This needs to be described more precisely. It would be of informative if the author could mention a biphasic susceptible age affecting young girls before starting menstruation and postmenopausal females.
- Basically, the direction of the introduction seems vague, as only gives a broad range background of the disease, and should rather be more relevant for PDT therapy. Also, I do not understand why the author provides too much information for VEGF, COX-2, and CDKN2A hypermethylation, all of these have not been investigated in this study, and instead, why the author did not exaggerate the potential treatment difficulty of LS, which may sometimes complicate the local malignancy in the clinical course (Lee A, JAMA Dermatol 2015;151:1061-7/ Cooper SM, et al. JAMA Dermatol 2015;151:1059-60). Also, the scenario for estrogen and its receptor has never been estimated in this study, without any of related descriptions to the PDT therapy. All the discussion is out of the perspective of this study.
- The patient’s details should be described more precisely; for example, i) what sort of local pharmacotherapy has been ineffective (Materials & Methods, 1st paragparh), ii) the study criteria for what the local therapy was ineffective (how long has it been administrated, and how much the major symptoms have been unhelpful, etc). All these details, including should be given as a table.
- The following details of the PDT procedure need to be at least given; i) the company, and trading name and number of photosensitizer, ii) how it has been dissolved into what solvent(s) and its final concentration to apply to the skin, iii) how many patients have shown the local hypersensitivity to the photosensitizer, and given up participating the study, otherwise regardless of whether patients showed the primary hypersensitivity they have been included in the study, etc.
- As shown in Fig 1, the group A’s patients included 34% vulval intraepithelial neoplasia. Have their neoplastic changes been developed in and/or next to the preceded LS lesion, and not yet been treated surgically? If so, they should not be integrated into the current study? It looks already beyond the study perspective. In addition to this, the group A also included a wide variety of malignancies, which were unclearly stated for the treatment intervention. Otherwise, it seems difficult to access to the clinical efficacy of PDT therapy alone?
- The QOL before and after PDT therapy has been estimated by 10-degree scale in each 3 major symptom, itching, burning, and pain (Table 3). How has the author traced to the 3 limited parameters into the % improvement in the satisfactory assessment; for example, if the patient with itching 10, burning 5, and pain 3 before the therapy turned to that with itching 5, burning 2, and pain 0, how do you categorize this patient to the grade what?
- If the group with less than 30% improvement after PDT therapy recognized as a poor responder (Table 3; D, E, N, and S), approximately 20% of the cohort showed unfavorable outcome. The author should discuss properly why such a large group was unresponsive to the PDT therapy
- It would be better if the author could give a table for symptomatic improvement, for example, total and each 3 group (3.1.2. Subjective benefits of PDT).
- The results from vulvoscopic assessment should be analysed by proper statistical assessments, to demonstrate the significance before and after PDT therapy between groups.
- The description for the average time of patient’s visit should be moved to material and method section (Discussion, 3rd paragraph).
Reviewer 3 Report
Here my major concerns
35-35
these epidemiological data must be contextualized. Data form ref 3 is related to Ducth population only.
53-54
One of the risk factors of lichen sclerosus is human papillomavirus virus (HPV), as 53
in cervical, vulvar and endometrial cancer [11]
Although some patients with lichen sclerous and / or HPV have endometrial cancer,
it does not seem to me that these factors are a relevant risk factor for endometrial cancer ...
54-55
8% of HPV-related lichen in women does not seem to me to be "high prevalence"
Overall, the introduction seems long and confusing to me, it should focus mainly on lichen sclerosus and its treatments
Dicussions
The role of corticosteroid therapy needs to be further developed, there are important data on the use of these treatments that cannot be ignored
e.g. doi:10.1001/jamadermatol.2020.1074